# Coping Styles in the Domestic Cat (*Felis silvestris catus*) and Implications for Cat Welfare

**DOI:** 10.3390/ani9060370

**Published:** 2019-06-18

**Authors:** Judith Stella, Candace Croney

**Affiliations:** 1United States Department of Agriculture-Animal and Plant Health Inspection Service Center for Animal Welfare, 625 Harrison St, West Lafayette, IN 47907, USA; 2Department of Comparative Pathobiology, Purdue University, 625 Harrison St, West Lafayette, IN 47907, USA; ccroney@purdue.edu

**Keywords:** coping style, personality, cats, welfare, behavior

## Abstract

**Simple Summary:**

It is important for cat caretakers to understand individual differences in response to stress. Identifying coping styles in cats may lead to improved health and welfare outcomes. In this study, we collected information from cat guardians pertaining to personality traits then singly-housed each cat for three days to mimic admittance to a veterinary hospital or shelter. Behavior was recorded hourly and response to approach of a familiar and unfamiliar person was assessed at the end of day 3. We found individual differences in the behavioral responses of cats to the acute stress of cage confinement. Additionally, guardian-rated personality traits agreed with the response of the cats when confined to a cage, suggesting that domestic cats have different coping styles. Identifying individual differences in response to stressful events or environments may provide caretakers with important information leading to improved welfare.

**Abstract:**

Identifying coping styles in cats may lead to improved health and welfare. The aims of this study were to (1) identify individual differences in response to acute confinement, and (2) to assess the predictability of guardian-rated personality traits on behavior. Adult cats (*n* = 55) were singly housed in enriched cages and behavioral observations were recorded for three days. On day 3, familiar and unfamiliar person approach tests were conducted. Fecal glucocorticoid metabolites (FGM) were quantified from voided samples. A questionnaire assessing personality traits and sickness behaviors was completed by each guardian. Analysis identified two clusters—cats in Cluster 1 (*n* = 22) were described as shy, calm, mellow, and timid; cats in Cluster 2 (*n* = 33) were described as active, playful, curious, and easygoing. Multilevel mixed-effects GLM revealed significant differences between the clusters including food intake (C1 > C2, *p* < 0.0001), affiliative/maintenance behaviors (C2 > C1, *p* < 0.0001), vocalization (C2 > C1, *p* < 0.0001), hide (C1 > C2, *p* < 0.0001), perch (C2 > C1, *p* < 0.0001), and latency to approach a familiar (C1 > C2, *p* < 0.0001) and unfamiliar (C1 > C2, *p* = 0.013) person. No statistically significant differences in FGM concentrations were identified (cluster *p* = 0.28; day *p* = 0.16, interaction *p* = 0.26). Guardian-rated personality traits agreed with the response of the cats when confined to a cage, suggesting that domestic cats have different coping styles. Identifying individual differences in response to stressful events or environments may provide caretakers with important information leading to improved welfare.

## 1. Introduction

Cats face many challenges to their welfare when housed in cages in veterinary hospitals, shelters, or laboratories. These include physical components of the housing environment, such as temperature, noise, cage size, hiding and perching opportunities (or lack thereof) as well as the social environment, including the quality of the human–cat interactions, presence of unfamiliar cats, or other animals such as dogs [1,2,3,4]. Assessing individual differences in response to stressors during confinement may aid caretakers in making decisions that meet cats’ individual needs, thereby enhancing their welfare.

A stressor is a stimulus that disrupts homeostasis through activation of the hypothalamic stress response system that may be physical (e.g., heat/cold), psychological (e.g., perceived threat), or social (e.g., aggressive interactions) and categorized as either acute or chronic [5]. In order to reestablish homeostasis, the individual will elicit an adaptive response that includes both physiological and behavioral components. Exposure to unpredictable and uncontrollable environments that are intense, prolonged, or novel and exceed the individual’s ability to cope will result in negative outcomes [6,7,8]. These may include decreased food intake, inhibition of exploratory and social behaviors, learned helplessness, stereotypies, and aberrant immune responses [9,10,11]. For cats, such stressors include loud or unfamiliar noises, novel and unfamiliar places and objects, and the approach of strangers (humans, cats, or other animals) into their personal space. As in other species, breed and individual differences in temperament or stress susceptibility [12,13], as well as past experiences [14], also influence cats’ responses to the environment [15,16]. 

The terms personality, temperament, coping styles, and behavioral syndromes have been used interchangeably to mean behaviors that are relatively consistent across time and context (although see [17] for a criticism). Studies have identified several dimensions of felid personality [18]. A comparative study of 45 personality traits across species that included the domestic cat (*Felis silvestris catus*), Scottish wildcat (*Felis silvestris grampia*), clouded leopard (*Neofelis nebulosa*), snow leopard (*Panthera uncia*), and African lion (*Panthera leo*) identified three distinct factors from a principal components analysis for each species [19]. In that study, 20 caretakers rated 100 domestic cats’ in two shelters and the factors identified were as follows: the factor Dominance had the highest loadings on the traits aggressive to conspecifics, bullying, and dominant; the factor Impulsiveness had the highest loadings on the traits excitable, active, and playful; and Neuroticism had the highest loadings on the traits, anxious, insecure, tense, suspicious, and fearful of people [19]. Litchfield et al. [20] assessed 52 personality traits of 2802 cats rated by their guardians and identified five factors that represent traits related to Neuroticism, Dominance, Impulsiveness, Agreeableness, and Extroversion while two recent studies both identified six dimensions [21,22].

While discrepancy in the number of dimensions identified, naming conventions, and methodology across studies makes it difficult to utilize the potential benefits of understanding feline personality, including it as part of a welfare assessment could lead to improved care and health outcomes for cats. For example, a relationship between immune response and personality has been reported [23,24]. Cats who fall on the bold end of the shy–bold continuum in relation to conspecific social behavior have been reported to be at increased risk for contracting feline immunodeficiency virus (FIV) [25]. In this study, we use the term coping style as defined by Koolhaas et al. [26] as “a coherent set of behavioral and physiological stress responses which is consistent over time”. Two distinct coping styles have been identified in diverse species including fish (e.g., [27]), red jungle fowl (e.g., [28]), rodents (e.g., [29]), dogs (e.g., [30]), and non-human primates (e.g., [31]). The proactive coping style is an active response characterized by high sympathetic nervous system (SNS) activation and low hypothalamic–pituitary–adrenal (HPA) axis activation, while the behavioral response involves territorial control and aggression. The reactive coping style is a withdrawal response and is characterized by higher HPA-axis activation, higher parasympathetic reactivity and behaviorally by immobility and low levels of aggression [26,32].

Quantifying the intensity and duration of the activation of the HPA-axis is an important aspect of coping style studies. One non-invasive tool that can be utilized is measuring fecal glucocorticoid metabolites (FGM) which will be present in feces after a species-specific time delay roughly corresponding to gut passage time [33]. In cats, radiometabolism studies have shown that approximately 82% of glucocorticoid metabolites are found in feces with peak concentrations approximately 22 h after administration [34]. Therefore, FGM analysis may provide a useful, non-invasive way to monitor stress responses in cats.

Identifying coping styles in cats in response to challenging environments may lead to improvements in health and welfare when confined. Therefore, the aims of this study were to (1) identify individual differences in response to acute cage confinement, and (2) to assess the predictability of those responses based on guardian-rated personality traits in the home environment. We hypothesized that cats that have a reactive coping style would have decreased food intake, eliminations, and vocalizations, spend more time hiding and have higher FGM concentrations than cats that have a proactive coping style, and that these responses would be associated with guardian-rated personality traits. 

## 2. Materials and Methods

### 2.1. Subjects

Adult cats were recruited from faculty, staff and students of Purdue University’s College of Veterinary Medicine. Fifty-five neutered, non-pedigreed cats (*n* = 28 female, *n* = 27 male) from 36 households completed the study (mean age 5.2 years, range 1–12.8 years). All cats were healthy and current on vaccinates for viral rhinotracheitis, calicivirus, panleukopenia, and rabies at the time of participation. Participants were admitted to the study between 16:00 and 19:00 h on day 0 and placed in an individual stainless-steel cage then released back to their guardians between 16:00 and 19:00 h on day 3. Guardians provided informed consent when their cat(s) were volunteered for the study. The Institutional Animal Care and Use Committee, Institutional Review Board, and the Clinical Trials Office in the College of Veterinary Medicine at Purdue University approved all experimental procedures (No. 1402001023).

### 2.2. Housing Environment

Cats were housed in the Purdue University vivarium in individual stainless-steel cages measuring 73 × 73 × 73 cm (Figure 1). The front half of the cage floor was covered by a mat (Sporttime yoga mat, School Specialty, Appleton, WI, USA). Each cage contained a cardboard box (ShelterDen by C Specialties Inc., Indianapolis, IN, USA) that had a hiding area (23 × 23 × 23 cm) with two access openings, placed in the rear corner of the cage. A shelf for perching (53 × 30 cm) was provided on the outside wall of the cage 27 cm from the top and bottom of the cage, flush with the back wall of the cage. Bedding (72 × 53 cm towel folded into quarters) was provided in the hide box and on the perch. A plastic litter pan (24 × 25 × 10 cm) was placed in the front half of the cage, filled with clumping litter (Tidy Cat, Nestle Purina Petcare Company, Wilkes Barre, PA, USA) at a depth of 3 cm. Guardians were asked to supply their cats’ regular diets and cats were fed per the instructions provided. Food and water were provided in separate 0.6 L stainless steel bowls. 

The housing room had dimensions of 57 m × 84 m and held 20 cages along two walls facing each other. A 13:11 (07:00: 20:00 h) light:dark schedule was maintained to approximate length of day in Indiana, USA, at that time of year (September/October). Room temperature (mean ± SD) of 22 ± 1.6 °C (72 ± 4°F) was maintained throughout the vivarium. Disturbances from people, barking dogs, or other unpredictable noises and events were avoided to the extent possible. Cages were spot cleaned by a single researcher to minimize disruption to the cat and ensure consistent handling. The study was conducted in three replicates with no more than 20 cats randomized to each replicate.

### 2.3. Data Collection

Prior to the cats’ arrival at the veterinary school, guardians completed a questionnaire on sickness behavior, personality traits (adapted from [19]), and management practices (reported in [35]) (Appendix A).

Prior to routine husbandry each day, one researcher stood in front of each cage for 30–60 s to record the previous night’s food intake, urination, defecation, and other sickness behaviors (e.g., vomit, diarrhea, or eliminations outside of the litter pan) for each cat (Appendix B). During husbandry, any evidence of the cat’s behavior or state that could not be seen from outside the cage was recorded (e.g., cached food, vomit, or eliminations outside of the litter pan). All husbandry procedures occurred between 08:00 and 10:00 each day.

Behavioral observations were collected between 08:00 and 16:00 h on day 1 and 2 and 08:00 and 14:00 on day 3 using two sampling techniques. A scan sample was collected every 2 h that included the cat’s position in the cage, the type of behavior(s) they were exhibiting, and vocalizations based on an ethogram for cats in cages (adapted from [2]) (Appendix C). The observer stood quietly in the middle of the housing room and recorded these parameters. Duration of observation time for each scan sample was approximately 3 min. Because observer effects on the behavior of the cats could not be ruled out during scan sampling, on the alternate hours, a five-minute, continuous focal sample using the same ethogram for cats in cages was video recorded for later coding. Two cats were recorded simultaneously for 5 min with a total of five replicates per observation hour so that each cat was recorded once per recording hour. Video cameras (Canon 53× HD Vixio HF R40, Canon USA, Inc., Melville, NY, USA) were placed on tripods, one recording a cat housed in an upper cage and one recording a cat housed in a lower cage, while the researcher left the housing room to reduce observer effects on the cats’ behavior. All videos were coded by one researcher.

After the last scan sample observation on day 3, all cage doors were covered, and a three-step unfamiliar person approach test followed by a familiar person approach test (adapted from [36]) was conducted starting approximately 30 min after cage doors were covered. Cats were tested in a randomized order for both tests. Each cat cage was uncovered for the test immediately prior to commencement of step 1. One woman, unfamiliar to the cats, served as the stranger and the caretaker served as the familiar person. Data were recorded live as well as video recorded for further analysis. During step 1, the person stood quietly 1 m from the cage for 30 s. In step 2, the person placed her hand on the cage door and stood quietly for 30 s. In the last step (step 3), the person opened the cage door and stood quietly with her hand extended toward the cat for 30 s. The cage door was re-covered immediately after completion of step 3. Latency to interact, duration of interaction, and the cat’s response to approach were recorded at each step. Response to approach was scored as follows: 1 = actively avoidant, aggressive, or displaying other signs of distress; 2 = avoidant without showing aggression or signs of distress; 3 = remaining in the same position in the cage; with or without purring, rubbing, kneading paws; 4 = responding positively to the person, approaching observer; 5 = actively seeking interaction with observer; rubbing cage door, rolling, purring, meowing, soliciting play, etc.

### 2.4. Fecal Glucocorticoid Metabolite (FGM) Analysis

Fecal samples were collected daily from the litter pan of each cat that defecated, individually placed into a sealed plastic bag, labeled and frozen at −20 °C until analysis. Samples were shipped overnight on dry ice to the Chicago Zoological Society Endocrinology Service Lab for analysis. Fecal glucocorticoid metabolites (FGM) were extracted using 80% ethanol in dH_2_O. First, 0.5 g (±0.05 g) of each fecal sample was weighed out (Mettler balance, model #AB104-5) into 16 × 125 mm polypropylene tubes. Next, 5 mL of 80% ethanol solution was added to each extraction tube. Each tube was vortexed and placed on a rotator (Labline Maxi Rotator, model #4631/Fisher) overnight (14–18 h). Each tube was then centrifuged for 15 min at 1500 rpm (Marathon 3000R centrifuge, model #120). For each sample, 1 mL of supernatant and 1 mL of assay buffer (0.1 M phosphate buffered saline containing 1% BSA, pH 7.0) was pipetted into 12 × 75 mm polypropylene tubes to produce a 1:10 dilution and frozen at −20 °C until assay analyses.

The samples were assayed using a commercially available corticosterone EIA kit (Enzo Life Sciences, Ann Arbor, MI, USA, catalog # 901-097), to determine FGM concentrations. The manufacturer supplied all needed instructions and components. Plates were read on a photospectrometer plate reader (Dynex MRX Revelation) at 405 nm. Biochemical validation of the assays consisted of the following: (1) parallelism with the standard curve, and (2) recovery to determine the percentage of exogenous hormone measured. For the recovery test, one sample was spiked with the highest five standards. The cross-reactivity of the Enzo Life Sciences corticosterone anti-body is as follows: 100% corticosterone, 28.6% desoxycorticosterone, 1.7% progesterone, 0.28% tetrahydrocorticosterone, 0.18% aldosterone, 0.13% testosterone and any other steroids were <0.05%. Assay sensitivity was 26.99 pg/mL and the intra-assay coefficient of variation was 6.63% at 44.79% binding. Inter-assay variation was 4.72% at 60.49% binding. The recovery of exogenous corticosterone (250–4000 pg/mL) was 143.58% in domestic cats. All hormone data are expressed in ng/g wet feces.

### 2.5. Statistical Analysis

#### 2.5.1. Cluster Analysis

Two cluster analyses were performed on information provided by the guardians prior to the study. The aim of this analysis was to determine if the reported sickness behavior and personality traits in the home correlated with those exhibited when the cat was confined, indicating a consistent coping style. Cluster analysis one (CA1) (kmeans, Gower measure for mixed data) grouped cats by the guardian-reported information that consisted of 33 variables related to personality traits (describes your cat, 1 = yes or 0 = no) and frequency of sickness behaviors (0 = I have never seen it, 1 = I have seen it at least once, 2 = I see it at least once per year, 3 = I see it at least once per month, 4 = I see it at least once per week, 5 = I see it daily) in the home environment. Fifty-one of the 55 cats in the study fit into one of the two clusters; Cluster 1 contained 62.8% of the cats (*n* = 32) and Cluster 2 contained 37.2% (*n* = 19) of the cats. Cluster analysis two (CA2) (kmeans, matched measure for binary data) grouped cats by the guardian-reported cat personality traits that consisted of 16 dichotomous variables. All 55 cats fit into one of the clusters; Cluster 1 contained 40% of cats (*n* = 22) and Cluster 2 contained 60% of the cats (*n* = 33).

#### 2.5.2. FGM

A 2-way ANOVA (cluster, day and their interaction) was performed on FGM concentrations.

#### 2.5.3. Multilevel Mixed-Effects Generalized Linear Models

Multilevel mixed-effects generalized linear models (Gaussian distribution, identity link function) were used to examine the association between cluster and the data collected in the vivarium environment. Four models were constructed, one each for sickness behavior, scan sampled behaviors, focal sampled behaviors, and approach test due to variation in the number of observation points for each type of data. Each model was run twice, once with CA1 as the outcome variable and once with CA2 as the outcome variable.

#### 2.5.4. Sickness Behavior (SB)

Food intake, urination, and defecation were the independent variables, cluster was the dependent variable and day (*n* = 3) was the random effect. Cage use and other SB were removed from analysis due to lack of variability and low frequency, respectively. 

#### 2.5.5. Scan Sampling

Behavior (affiliative, aggressive, agonistic), position in cage (front half, rear half, in litter pan, in hide box, perching) and vocalizations (none, meow, hiss/growl) collected by scan sampling were the independent variables, cluster was the dependent variable and observation (14 per cat) was the random effect. 

#### 2.5.6. Focal Sampling

Duration of observation time spent occupied in 12 behaviors were included in the model; position in cage (front half, rear half, hide box, perch), alert relaxed, rest, groom, alert tense, freeze, eat, attempt to hide, turn away. The other variables on the ethogram were removed due to infrequency of observation (<10%). Cluster was the dependent variable and observation (12 per cat) was the random effect.

#### 2.5.7. Approach Test

Latency to approach a familiar and an unfamiliar person and response to approach of a familiar and an unfamiliar person were included as independent variables, cluster was the dependent variable and step of the test (*n* = 3) was the random effect. Duration of interaction with the familiar and unfamiliar person were removed from the analysis, as they were the inverse of latency to approach for the majority of cats.

All analyses were performed using STATA 15 (StataCorp LP, College Station, TX, USA) and graphed in GraphPad Prism 7 (GraphPad Software Inc., La Jolla, CA, USA).

## 3. Results

### 3.1. Cluster Analysis 

CA1: Grouping cats using guardian-reported sickness behavior and personality traits failed to predict response to the confinement experience. The only statistically significant result was food intake (coefficient −0.0969, 95% CI −0.1846, −0.00092, *p* = 0.003). The remainder of the paper will discuss the results of the models using clusters from CA2.

CA2: Table 1 presents the percentage of guardians reporting each personality trait in their cat. Cats in Cluster 1 were more likely to be described as shy, calm, mellow, and timid with strangers, while cats in Cluster 2 were more likely to be described as active, playful with people, social with strangers, curious, easygoing, needy, and social with familiar people. 

The percentage of cats reported to exhibit sickness behaviors at least once per month is presented in Table 2. Cats in Cluster 2 were more likely to vomit, defecate outside the litter pan, groom excessively, and exhibit fearful, nervous, and aggressive behaviors than cats in Cluster 1.

### 3.2. Fecal Glucocorticoid Metabolite

No statistically significant differences were identified (cluster *p* = 0.28; day *p* = 0.16, interaction *p* = 0.26) by 2-way ANOVA although a trend was found of increased FGM concentrations for cats in Cluster 1 on days 2 and 3 compared to cats in Cluster 2 (Table 3, Figure 2).

### 3.3. Sickness Behavior (SB) 

The percentages of cats eating greater than half of the offered food (*p* < 0.0001) and urinating in the litter pan at least once per day (*p* < 0.0001) were statistically significant. Cats in Cluster 1 were more likely to eat greater than half the offered food on days 2 and 3 and more likely to urinate in the litter pan on day 1 than cats in Cluster 2 (Table 4).

### 3.4. Scan-Sampled Behavior 

Cats in Cluster 2 were more likely to exhibit affiliative or maintenance behaviors (*p* < 0.0001) and meow (*p* < 0.0001) during the study and did so earlier than cats in Cluster 1 (Table 4). Cats in Cluster 1 spent more time in the hide box while cats in Cluster 2 were more likely to be observed perching throughout the study (Table 4, Figure 3). 

### 3.5. Focal Sampling Behavior 

Cats in Cluster 1 spent more of the observation time in the front half of the cage and in the hide box than cats in Cluster 2 while cats in Cluster 2 spent the majority of their time perching (Table 4). Cats in Cluster 2 spent more of the observation time alert relaxed or resting while cats in Cluster 1 spent more time alert tense or freezing (Table 4).

### 3.6. Approach Test 

Cats in Cluster 2 had a shorter latency to approach a familiar person (*p* < 0.0001), an unfamiliar person (*p* = 0.013) and showed more affiliative responses to approach by an unfamiliar person (*p* < 0.0001) than cats in Cluster 1 (Table 4, Figure 4).

## 4. Discussion

The guardian rating, behavioral coding, and physiologic stress response (FGM) data collected in this study suggest that individual differences in response to acute confinement in a cage exist and that guardian-reported cat personality traits were consistent with their observed behaviors when confined. Behavioral and physiologic response patterns suggest that cats in Cluster 1 had a reactive coping style while cats in Cluster 2 had a proactive coping style. Koolhaas and van Reenan [37] have proposed a 3-dimensional model of response patterns to stressors with coping style, emotionality, and sociality as independent dimensions. Coping style is a behavioral pattern that reflects the type of response an individual makes (qualitative) while emotionality is the intensity of the response reflected in the duration of the behavior and physiologic activation (quantitative). The third dimension, sociality, has been proposed as a separate trait that appears to be largely uncorrelated with other behavioral and physiological measures. Individuals with a proactive coping style tend to be less behaviorally flexible relying on previous experience to influence their responses while individuals with a reactive coping style are more flexible and cue-dependent, continually gathering information from the environment and then reacting. Therefore, individuals with a proactive coping style are more successful under stable conditions while those with a reactive coping style are more successful in variable conditions [26,32].

Two methods are commonly used when assessing animal personality or coping style: coding and rating. The rating method involves an observer making judgments about an individual animal’s behavioral traits based on experience with that individual and is increasingly used in large-scale internet-based studies of companion animals (e.g., [38]). The coding method involves researchers observing an animal’s behaviors and describing it in terms of personality traits (e.g., [30]). In this study, both methods were employed as it has been suggested that a combined approach may provide more information, increasing the validity and reliability of the measures [39,40,41].

The guardian-rated personality traits identified two groups—cats in Cluster 1 were rated as shy, calm, mellow, quiet, and timid with unfamiliar people whereas cats in Cluster 2 were rated as active, curious, independent, and social with both familiar and unfamiliar people. Behavioral coding during the confinement stressor largely agreed with the guardian-ratings. When challenged with confinement stress, the cats in Cluster 1 exhibited freezing behavior, were alert and tense, and spent much of the time in their hiding box. In contrast, the cats in Cluster 2 were resting or alert and relaxed and spent much of the time perching. Gourkow et al. [42] reported similar behavioral differences in response to confinement of 34 cats admitted to a shelter. Three dimensions of behavior were identified: 1 included behaviors such as hiding, freeze, flat, startle, retreat from humans; 2 included meow, scan, escape bouts, pacing, redirected aggression, and 3 included lie on side, sleep, friendly to humans, walk, eat, groom, and rub. The behaviors included in Dimension 1 appear to be similar to those of our Cluster 1 while Dimension 3 resembles our Cluster 2. We could not identify a group corresponding to Dimension 2.

In this study, we provided a housing environment that was quiet, predictable, and enriched, to minimize stress and promote acclimation [2,43]. Cats had the opportunity to hide and perch and the two clusters utilized the resources differently. While hiding behavior decreased and perching behavior increased across days in both groups, there was still a defined preference for hiding or perching based on cluster with cats in Cluster 1 preferring to hide and cats in Cluster 2 to perch. This suggests the importance of supporting individual animals’ needs by providing both hiding and perching opportunities to confined cats, as has been previously reported [2,43,44,45,46]. 

No statistically significant difference in FGM concentrations was identified. This is likely due to the infrequency of defecation; only eight cats voided on each day of the study. However, cats in Cluster 1 showed a trend toward increased FGM concentration on days 2 and 3 compared with day 1, consistent with a reactive coping style (Figure 2). Cats in Cluster 2 had a relatively constant FGM concentration across days that was lower than the cats in Cluster 1, consistent with a proactive coping style. FGM concentrations are considered a reliable metric of HPA-axis activation in cats, with the major route of glucocorticoid metabolite excretion being via feces [33]. Ellis et al. [47] found FGM concentrations to be elevated during week 1 gradually decreasing by week 5 in singly-caged cats studied over a 30-day period. The mean FGM was lower in week 1 than was observed in our current study, possibly due to differences in reporting; Ellis et al. [47] reported a weekly average whereas we assessed concentration daily. The current study may have captured the peak response, suggesting that all cats experienced stress during confinement and may not have fully acclimated by the end of the study. Affiliative and maintenance behavior, food intake and urination in the litter pan increased in all cats from day 1 to day 3 following the acclimation pattern previously reported [2,43]. It is possible that FGM was beginning to decrease as well and that a longer study duration might have indicated this given that Ellis et al. [47] reported a significant decrease in stress behavior from week 1 to week 2 while FGM concentrations took longer to return to baseline. Future research is needed to assess cats’ FGM responses during confinement and to determine if the trends identified here are consistent in a larger study population.

We also identified differences in cat sociability and found them to be consistent between the guardian-reported ratings and the coded behavioral observations. Guardians described cats in Cluster 1 as timid with strangers but friendly with familiar people. During confinement, they were slower to exhibit affiliative behavior, spent much of the time in their hide box, and in the approach test had a longer latency to approach the unfamiliar person compared to cats in Cluster 2. In contrast, cats in Cluster 2 were reported to be social with both familiar and unfamiliar people. When confined, they exhibited affiliative behavior early and throughout the study and meowed more often. Cat vocalizations are more common in the presence of a human and meowing is an intra-specific attention-seeking behavior [48,49] supporting the increased sociality of this cluster. Interestingly, during the approach test, the average latency to approach the familiar person was similar in both clusters (Figure 4a). However, cats in Cluster 1 had a longer latency and exhibited fewer affiliative behaviors in response to approach of an unfamiliar person (Figure 4b,d). There was high variability between individuals in both clusters, possibly supporting Koolhaas and van Reenan’s [37] third dimension of sociality. Regardless, this finding demonstrates the importance of consistent caretakers and suggests that development of a human–cat relationship may be formed in as little as 3 days, even for less sociable cats. 

Differences in owner-reported sickness behavior between clusters were identified. Cats in Cluster 2 were more likely to vomit, defecate outside the litter pan, groom excessively, and exhibit fearful, nervous, and aggressive behaviors than cats in Cluster 1. Sickness behaviors have previously been reported to increase in response to unusual external events [10,11] in a laboratory environment. Increased aggression and defecating out of the litter box may be indicative of territorial behaviors consistent with a proactive coping style, as has been reported in rodents [32]. It is less clear why vomiting, excessive grooming, and fearful and nervous behaviors would be increased, but one explanation is that these cats may be less behaviorally flexible to changes in their environment, and therefore may experience more stress, which manifests in these behaviors. Future research should aim to look for an association between the predictability of the home environment and sickness behavior.

This study had several limitations. First, the tool used by guardians to rate personality traits was not validated and data were collected as a dichotomous variable, whereas a 5-point Likert scale of each trait is more often utilized. Even with this limitation, though, individual differences were identified. Second, we coded behavior only in one context—cage confinement. Ideally, many contexts across time should be assessed to ascertain consistent individual responses. Finally, the sample size was small and since all cats were volunteered, volunteer bias may have been introduced. 

Recognizing how individuals respond to stressful events or environments may help caretakers make informed, welfare-friendly decisions pertaining to cat housing, handling, enrichment, and resource allocation. This has the potential to improve cat welfare, facilitating better physical health, shorter times to adoption, and improved human–cat interactions. Future studies should aim to develop a validated personality assessment for cats and expand this investigation with a larger sample size in multiple contexts.

## Figures and Tables

**Figure 1 animals-09-00370-f001:**
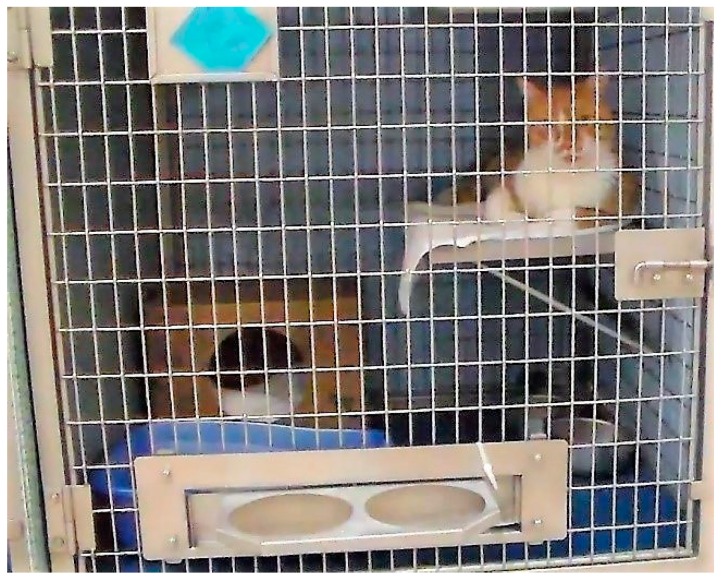
Study cat during confinement showing cage set-up.

**Figure 2 animals-09-00370-f002:**
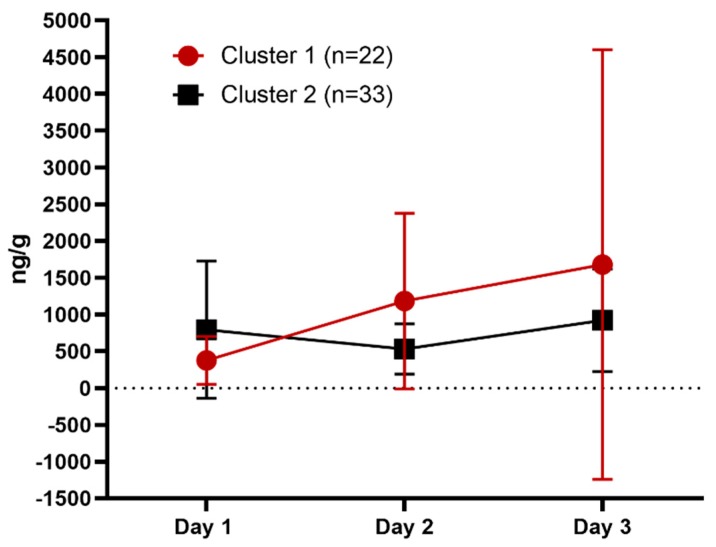
Fecal glucocorticoid metabolite (FGM) concentrations (ng/g) by cluster and day (mean and standard deviation).

**Figure 3 animals-09-00370-f003:**
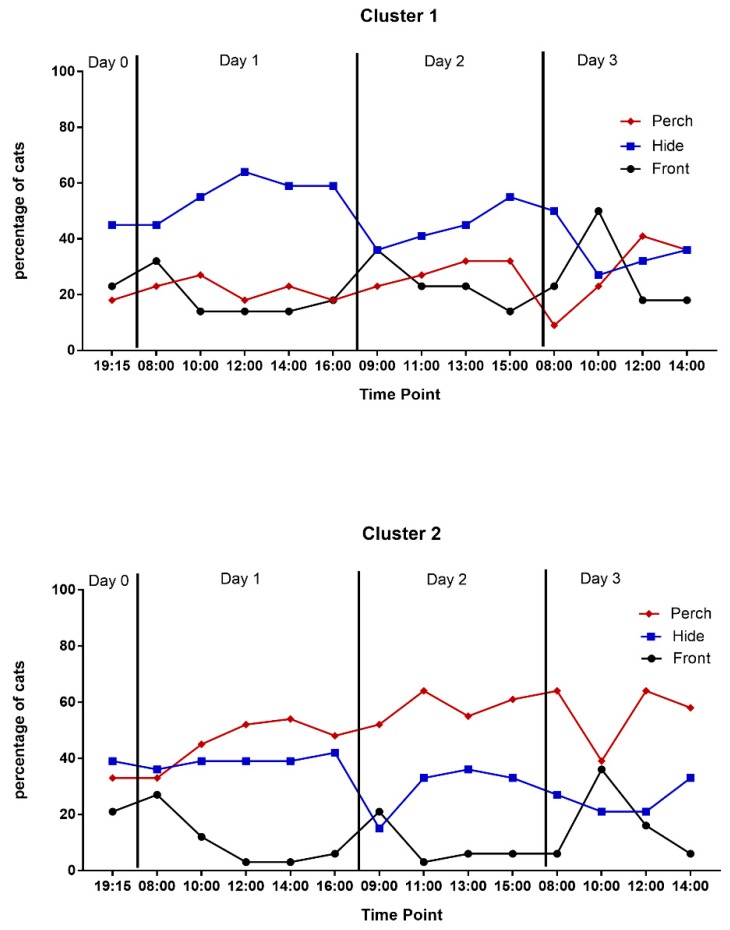
Percentage of cats in the front half, hide box, or perch by cluster during scan sampling.

**Figure 4 animals-09-00370-f004:**
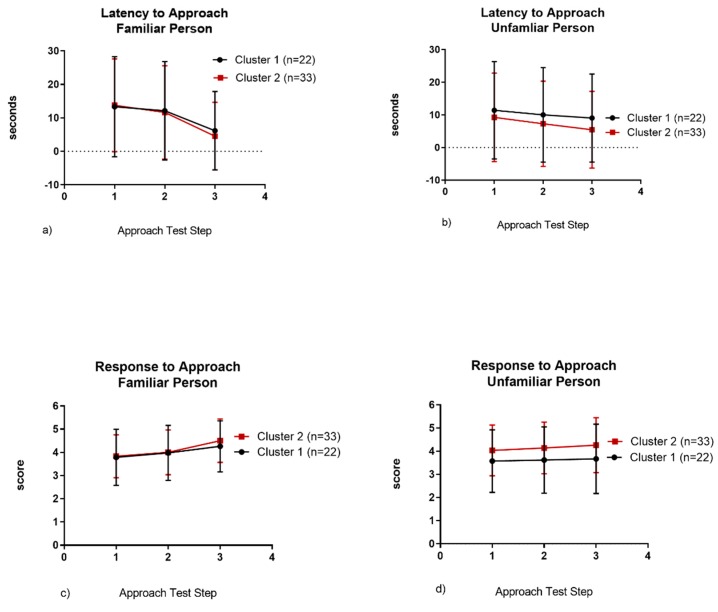
Latency to approach a familiar person (**a**) or unfamiliar person (**b**) and response to approach of a familiar (**c**) or unfamiliar person (**d**).

**Table 1 animals-09-00370-t001:** Percentage of cats rated by their guardians with each personality trait.

Personality Trait	Cluster 1 (*n* = 22)	Cluster 2 (*n* = 33)
Active	9	89
Shy	55	6
Calm	41	27
Mellow	41	27
Playful with people	5	76
Vocal	36	48
Likes attention but is not demanding	36	48
Social with strangers	23	67
Curious	5	91
Independent	55	73
Quiet	41	15
Easygoing	27	70
Playful with toys	50	76
Needy, forces attention	18	42
Social with familiar people	41	79
Timid with strangers	59	21

**Table 2 animals-09-00370-t002:** Guardian-reported frequency of sickness behaviors by cluster (% reporting at least 1/month).

How Often Does Your Cat:	Cluster 1 (*n* = 22)	Cluster 2 (*n* = 33)
Have excessive appetite	41	36
Have little appetite	5	0
Vomit (food, hair, bile, other)	14	39
Have diarrhea	0	3
Have constipation	0	1
Defecate outside the litter pan	0	12
Strain or have frequent attempts to urinate	0	0
Urinate outside the litter box	9	9
Have blood in the urine	0	0
Spray urine	5	6
Grooms excessively	0	9
Have excessive hair loss	5	3
Scratch themselves excessively	5	3
Have discharge from eyes	9	3
Seem nervous (anxious)	5	27
Seem fearful	18	30
Seem aggressive	9	15
Seem “needy” of contact or attention	68	67

**Table 3 animals-09-00370-t003:** Fecal glucocorticoid metabolite (FGM) concentration (ng/g) by cluster and day.

Day	Cluster	N (%)	Mean	SD	Min	Max
1	1	7 (32)	375	325	159	1054
	2	18 (55)	792	934	9	3451
2	1	11 (50)	1185	1195	123	3757
	2	15 (45)	531	341	148	1300
3	1	11 (50)	1679	2923	93	1061
	2	19 (58)	922	698	174	2130

**Table 4 animals-09-00370-t004:** Results of a mixed effects generalized linear model (*n* = 55). FP familiar person; UFP unfamiliar person.

Predictor	Coefficient	95% CI	*p*-Value
Sickness behaviors			
Food Intake	0.1039	0.07, 0.14	<0.0001
Urination	−0.1798	−0.22, −0.14	<0.0001
Defecation	−0.0735	−0.1799, 0.0329	0.18
Scan sampled behaviors			
Behavior	−0.0882	−0.1261, −0.0503	<0.0001
Position	0.0533	0.0357, 0.0709	<0.0001
Vocalizations	0.1689	0.1192, 0.2187	<0.0001
Focal sampled behaviors			
Front half	−0.0022	−0.0026, −0.0018	<0.0001
Rear half	−0.0017	−0.0022, −0.0013	<0.0001
Hide	−0.0014	−0.0018, −0.0009	<0.0001
Perch	−0.0008	−0.0011, −0.0004	<0.0001
Alert relaxed	−0.0007	−0.0015, 0.00002	0.06
Rest	−0.0011	−0.0019, −0.0003	0.009
Groom	−0.00002	−0.0008, 0.00077	0.9
Eat	0.002	0.0001, 0.0023	0.03
Alert tense	−0.0011	−0.0019, 0.0002	0.02
Freeze	−0.0021	−0.0028, −0.0014	<0.0001
Attempt to hide	−0.002	−0.0026, −0.0015	<0.0001
Turned away	−0.0002	−0.0007, 0.0004	0.5
Approach tests			
Latency to approach FP	0.0037	0.0035, 0.0039	<0.0001
Latency to approach UFP	0.0046	0.0009, 0.0082	0.013
Response to approach FP	−0.0297	−0.0853, 0.0258	0.3
Response to approach UFP	0.1529	0.0858, 0.2199	<0.0001

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
