# Peer review of "Coping Styles in the Domestic Cat (Felis silvestris catus) and Implications for Cat Welfare"

_animals, 2019, doi:10.3390/ani9060370_

Round 1

Reviewer 1 Report

L38-39: I feel that this is far more relevant to clinics than to owners, the owner can't really manage care at clinic if the cat needs to be there (e.g. for a surgical procedure).

I feel it is also important to acknowledge that a cat would rarely be kept in this manner in a clinic were it not also experiencing illness/recovery. These aspects could substantially alter expected behaviour and coping-style.

It is also hard to extrapolate the findings to non-companion cats (as this sample was solely taken from that sub-population), therefore including laboratories in the wider implications could be erroneous.

L61: to the [what]?

L63: although see MacKay and Haskell (this journal) for a criticism as to whether these are actually the same thing.

L79 "has been reported"

In the final paragraphs of the intro we shift to the terms reactive and proactive. It is important to use the same terminology throughout, and make sure in the abstract that the "clusters" are ascribed to these terms.

L138: the extent that it was possible (or "where possible")

L143: given the assertion earlier in the introduction that so many different tools have been used that it is difficult to accurately amalgamate personality types I wonder why the authors chose to further adapt a tool.

L151: I would encourage the authors to simply say exactly what observations were made and when. They can then expand on the nuances (cameras; observer position etc.) in the next paragraphs. The crux of the timings (e.g. scan per 2h) is lost amongst the wider methods.

Given that focal sampling will automatically provide more granularity than a scan sample I wonder why there is a model constructed solely for the scan sampling.

Would it be worth looking at cats that did not defecate and personality? There may be a sig diff between the 2 clusters, and this may explain why the FGM was NS.. 

Fig 2 needs the error bars, these will identify why they are NS (due to, one assumes, substantial overlap)

There is no mention of how much food was provided. If this was not standardised (in grams) then it should not be included. The amount eaten will be related to the amount offered

.Table 4 needs to identify which cluster is used as the comparator.

Fig 4 c/d this is a behavioural response, so why is the y-axis in seconds? This doesn't seem to make sense.

I can't really see how Fig 4a provides such a strong significant difference between clusters.. in fact, based on the visual presentation i would say there appears to be no sig diff.

L351: is this absence likely to be a result of the 2 groups assessed? (Cats entering a shelter may be trapped, or relinquished).

How does perching vs hiding relate to differences in reactivity or proactivity? I can't think of one, it would seem more related to traits than coping styles. Throughout the consideration of coping styles would seem less convincing than relationships with personality traits (which are not necessarily interchangeable)

L406: Data were collected as dichotomous variables (please treat 'data' as a plural noun throughout)

Author Response

Dear reviewer,

Thank you for taking the time to review this manuscript and provide thoughtful comments. See below for our responses.

L38-39: I feel that this is far more relevant to clinics than to owners, the owner can't really manage care at clinic if the cat needs to be there (e.g. for a surgical procedure).

We did not address this to a specific group (owners) but rather broadly to all cat caretakers. Additionally, while cat owners may not be able to care for their cat in a clinic, if they are provided with this information they would be able to advocate and/or select veterinary hospitals or boarding facilities that provide more cat welfare friendly services and practices.

I feel it is also important to acknowledge that a cat would rarely be kept in this manner in a clinic were it not also experiencing illness/recovery. These aspects could substantially alter expected behaviour and coping-style.

We based the housing model on our experience of veterinary and shelter housing. Additionally, coping style by definition is “a coherent set of behavioral and physiological stress responses which is consistent over time”. Therefore, if a cat has a reactive coping style they will be motivated to hide, and likely more so, when ill or recovering. If anything, we feel it will be more important to successful recovery.

It is also hard to extrapolate the findings to non-companion cats (as this sample was solely taken from that sub-population), therefore including laboratories in the wider implications could be erroneous.

We respectfully disagree. The cats in this study were originally obtained from a variety of sources so they had diverse backgrounds and experiences. Additionally, some cats used in biomedical research were previously companion animals. Finally, coping style is primarily a trait of the individual. While the ratio of proactive to reactive coping styles within a population may differ, the individual cat’s response when confined should be the same. Additionally, we have previously reported many behavioral similarities between laboratory housed cats and “owned” cats. (e.g.Stella et al., 2011, 2012).

L61: to the [what]?

Edited added ‘environment’

L63: although see MacKay and Haskell (this journal) for a criticism as to whether these are actually the same thing.

Added reference

L79 "has been reported"

edited

In the final paragraphs of the intro we shift to the terms reactive and proactive. It is important to use the same terminology throughout, and make sure in the abstract that the "clusters" are ascribed to these terms.

We did not shift to this terminology, this is the hypothesis we are investigating, We have maintained the clusters as 1 and 2 since we can not definitively state that these cats have reactive or proactive coping style, only that the data suggest this may be true.

L138: the extent that it was possible (or "where possible")

Left as written.

L143: given the assertion earlier in the introduction that so many different tools have been used that it is difficult to accurately amalgamate personality types I wonder why the authors chose to further adapt a tool.

This study was conducted in 2014 at which time many of the studies referenced had not been published yet. We adapted the personality scale from Gartner et al., 2014 prior to that paper being published based on personal communication with the first author. We acknowledge this is a limitation of the study and the field needs further research before these findings can be incorporated into welfare assessments.

L151: I would encourage the authors to simply say exactly what observations were made and when. They can then expand on the nuances (cameras; observer position etc.) in the next paragraphs. The crux of the timings (e.g. scan per 2h) is lost amongst the wider methods.

Left as written to maintain consistency with previously published studies using this study design and methodology (see Stella et al., 2012, 2014).

Given that focal sampling will automatically provide more granularity than a scan sample I wonder why there is a model constructed solely for the scan sampling.

Separate models were constructed due to the variation in the number of observation points as stated in lines 226-227. We incorporate scan sampling into the study because it is much more practical than focal sampling so if it provides the same information we can recommend it for daily observations of caged cat welfare.

Would it be worth looking at cats that did not defecate and personality? There may be a sig diff between the 2 clusters, and this may explain why the FGM was NS.. 

Unfortunately, this was not possible. Every cat defecated but only 8 did so on each of the 3 study days. For the others, there was no consistent pattern, some did on day 1 and 3, some on day 2 and 3 some on day 3 only or day 2 only.  Unfortunately, it is a limitation to this metric over a short time period and likely why others (see Ellis et al) reported weekly averages of FGM.

Fig 2 needs the error bars, these will identify why they are NS (due to, one assumes, substantial overlap)

Edited

There is no mention of how much food was provided. If this was not standardised (in grams) then it should not be included. The amount eaten will be related to the amount offered

We acknowledge your point, but the amount offered was the “normal” amount provided to each individual cat by the owner so we believe it is a relevant metric. Since most cats are of a relatively comparable size, the amount fed was not wildly different as would be expected with different dog breeds.

Table 4 needs to identify which cluster is used as the comparator.

Please clarify what you mean as it is not clear to us. Cluster was the dependent variable. A comparator would be relevant for independent variables if any were ordinal or interval but that was not the case in this study.

Fig 4 c/d this is a behavioural response, so why is the y-axis in seconds? This doesn't seem to make sense.

Graphs were accidentally mislabeled. Edited

I can't really see how Fig 4a provides such a strong significant difference between clusters.. in fact, based on the visual presentation i would say there appears to be no sig diff.

The difference is statistically significant although we agree that is not likely to be clinically significant. Future research should aim to build on these results to further address some of the limitations as stated in lines 413-415.

L351: is this absence likely to be a result of the 2 groups assessed? (Cats entering a shelter may be trapped, or relinquished).

We believe it is likely due to methodology and statistical analysis used. There is no consistent number of dimensions in personality studies either. This is a relatively new field requiring  more research to reach a consensus.

How does perching vs hiding relate to differences in reactivity or proactivity? I can't think of one, it would seem more related to traits than coping styles. Throughout the consideration of coping styles would seem less convincing than relationships with personality traits (which are not necessarily interchangeable)

A reactive coping style is characterized behaviorally by a withdrawal response so it is likely that a cat with this coping style will have an increased motivation to hide. In contrast, a proactive coping style is characterized behaviorally by territorial and aggressive behavior so these cats may be motivated to elevate for better observation of their surroundings. But even if it can’t be explained the fact that this difference appears to exist suggests that caretakers should provide both hiding and perching opportunities to allow the cat to respond as they are motivated to do, thereby improving their welfare.

L406: Data were collected as dichotomous variables (please treat 'data' as a plural noun throughout)

 edited

Reviewer 2 Report

This is a very well designed and executed study, that is also well written. I believe the subject matter is novel and the result has important potential to support strategies for helping cats cope with periods of acute stress when confined in unfamiliar environments.

In my opinion the research model, which involves housing domestic cats in a very confined space with little ability to express their natural behaviours, could be considered a moderately harmful event for those cats involved (these are cats that have been volunteered to take part in the study by their owner). Therefore the benefits of the study have to be justified. I believe the information obtained does justify the use of this model and the conclusions reached i.e. "Recognising how individuals respond to stressful events or environments may help caretakers make informed, welfare friendly decisions pertaining to cat housing." Ideally I would like to see this work followed up with practical information for caretakers about how best to use this research and the tools used so that the benefits are realised.

Identifying the two clusters was central to the success of this study. There is a risk that cats in Cluster 2 could be perceived by caretakers as less stressed than those exhibiting the behaviours of those cats in Cluster 1. The only physiological marker of stress measured in this study was FGM in faeces, and the data did not show any difference between the two groups of cats. Since only 8 cats voided on each day, it is not possible to identify a within cat adaptation to the stressful situation. A minor study improvement could have been to collect a faecal sample from the home environment to establish a baseline under normal living conditions.

One minor correction is required and that is completion of the sentence in line 61.

Author Response

Dear Reviewer,

Thank you for taking the time to review this manuscript. See out responses below.

This is a very well designed and executed study, that is also well written. I believe the subject matter is novel and the result has important potential to support strategies for helping cats cope with periods of acute stress when confined in unfamiliar environments.

In my opinion the research model, which involves housing domestic cats in a very confined space with little ability to express their natural behaviours, could be considered a moderately harmful event for those cats involved (these are cats that have been volunteered to take part in the study by their owner). Therefore the benefits of the study have to be justified. I believe the information obtained does justify the use of this model and the conclusions reached i.e. "Recognising how individuals respond to stressful events or environments may help caretakers make informed, welfare friendly decisions pertaining to cat housing." Ideally I would like to see this work followed up with practical information for caretakers about how best to use this research and the tools used so that the benefits are realised.

Identifying the two clusters was central to the success of this study. There is a risk that cats in Cluster 2 could be perceived by caretakers as less stressed than those exhibiting the behaviours of those cats in Cluster 1. The only physiological marker of stress measured in this study was FGM in faeces, and the data did not show any difference between the two groups of cats. Since only 8 cats voided on each day, it is not possible to identify a within cat adaptation to the stressful situation. A minor study improvement could have been to collect a faecal sample from the home environment to establish a baseline under normal living conditions.

Thank you for your comments. The authors agree that a baseline sample voided in the home environment would have been optimal and should be included in future studies.

One minor correction is required and that is completion of the sentence in line 61.

Edited added ‘environment’